# Algorithm of FBG Spectrum Distortion Correction for Optical Spectra Analyzers with CCD Elements

**DOI:** 10.3390/s21082817

**Published:** 2021-04-16

**Authors:** Vladimir Anfinogentov, Kamil Karimov, Artem Kuznetsov, Oleg G. Morozov, Ilnur Nureev, Airat Sakhabutdinov, Konstantin Lipatnikov, Safaa M. R. H. Hussein, Mustafa H. Ali

**Affiliations:** 1Department of Radiophotonics and Microwave Technologies, Kazan National Research Technical University Named after A.N. Tupolev-KAI, K. Marx Str. 10, 420111 Kazan, Russia; v.anfinogentov@yandex.ru (V.A.); mail12kamil2000@mail.ru (K.K.); AAKuznetsov@kai.ru (A.K.); microoil@mail.ru (O.G.M.); n2i2@mail.ru (I.N.); klipatnikov87@mail.ru (K.L.); 2Department of physics, College of Education for Pure Sciences, University of Karbala, Karbala 56001, Iraq; safaa.mohammed@uokerbala.edu.iq; 3College of Dentistry, University of Mustansiriyah, Baghdad 14022, Iraq; mustafa.h@uomustansiriyah.edu.iq

**Keywords:** nonlinear spectrum distortions, signal exposition time, fiber Bragg grating, fiber Bragg sensors, fiber optic sensors, fiber optic interrogator, optical spectrum analyzer, charge-coupled device elements, CCD

## Abstract

Nonlinear spectrum distortions are caused by the peculiarities of the operation of charge-coupled device elements (CCD), in which the signal exposition time (Time of INTegration–TINT) is one of the significant parameters. A change of TINT on a CCD leads to a nonlinear distortion of the resulting spectrum. A nonlinear distortion of the spectrum, in turn, leads to errors in determining the central wavelength of fiber Bragg gratings (FBGs) and spectrally sensitive sensors, which, in general, negatively affects the accuracy of the measuring systems. This paper proposes an algorithm for correcting the nonlinear distortions of the spectrum obtained on a spectrum analyzer using CCD as a receiver. It is shown that preliminary calibration of the optical spectrum analyzer with subsequent mathematical processing of the signal makes it possible to make corrections in the resulting spectrum, thereby leveling the errors caused by measurements at different TINT.

## 1. Introduction

It is widely known that in recent years, fiber-optic measuring systems have become increasingly relevant. In fiber-optic sensor systems, various technologies of interrogation and multiplexing are used [1,2,3,4,5]. Different technologies are applied for dividing fiber-optic spectral-sensitive sensors: by wavelength [1], by time response [2], by frequency [3], by polarization [4], and by spatial [5] multiplexing. To determine the average wavelength of the sensors, optical analyzers, such as spectrometers with tunable Fabry-Perot system interferometers, or diffraction gratings with CCD arrays, are used. The complexity of multiplexing technologies is also related to the fact that any spectrum overlaps of FBG spectra leads to significant errors in measurements of their central wavelengths [6,7,8]. The multiplexing technologies and microwave-photon interrogation methods of spectrally-encoded and addressed Bragg gratings have been developed; they allow separating, spectrally, the responses of sensors in the same frequency range [9,10,11,12]. Some researchers worked on sensor detection and tracking using Slepyan codes [13,14,15,16]; thus, a measurement of temperature and deformation in the case of sensors spectra overlapping became possible. Recently, a more convenient technology, based on addressable fiber Bragg structures, was proposed [17]. It made it possible to design distributed sensor systems with a large number of sensors without complicated optoelectronic schemes [17,18,19].

Despite significant progress in this direction, the classical optical spectrum analyzers, using diffraction gratings with CCD, have yet to develop their full potential. The attractiveness is in the lower cost of such devices, compared to interrogators on tunable filters (with comparable measurement errors). The source of optical radiation is superluminescent laser diodes with a spectral width that overlaps the working spectral range of the analyzer.

The information parameter of the FBG sensor is the shift of its central wavelength. The data obtained from the CCD array allow measurements with an accuracy of ~160 pm, which is not acceptable. Therefore, to improve the accuracy, an approximation is used. Various types of approximation have been investigated: Gaussian curve; second-order parabola using three upper points [20]; a parabola using the least-squares method; approximating the position of the central wavelength using the center of mass method; and others [21,22]. All investigated methods make it possible to determine the central wavelength FBG with a margin of error; however, the center of mass method gives the best accuracy.

## 2. Problem Explanation

For an experimental study of the FBG sensor interrogating system, its prototype was assembled. The IBSEN I-MON 512 USB as a spectrometer, a SLD-761-HP1-DIL as a broadband light source, and an FS62WSS (HBM) as a bore-type temperature sensor were used. The sensor response was measured at several constant temperature values maintained by a thermostat with an accuracy of ±0.1 °C. The dependence “shift of the central wavelength-temperature” was plotted based on the measurement results. The second series of measurements were made at the facility, where the sensor was connected to the device through an existing long optical line. The decrease in the optical signal power was compensated by the increase in the exposition time (TINT) of the CCD array, which made it possible to “scale” the spectrum to the required signal-to-noise ratio. The spectrometer manufacturer (IBSEN) uses the term TINT as the variable and command in the software, and the term means “Time of INTegration”. However, this approach led to the fact that the previously obtained correspondence “central wavelength–temperature shift” began to be violated: at the same temperature, the value of the central wavelength differed by tens of picometers (equivalent to an error of several degrees Celsius), from which a hypothesis was put forward about the nonlinear deformation of the spectrum with a change in the accumulation time.

It should be noted that there is no mention of this phenomenon in the user’s manual of the spectrometer, which is, in our opinion, due to the fact that the manufacturer considers the spectrometer as an end product; in this regard, its use as an element of a more complex system is not within their goals.

In the works of other authors, it is shown that CCD-based spectrometers have a number of spectrum distortions, such as nonlinear pixel sensitivity to incident light [23] and chromatic distortions [24,25,26]. In these works, the sources of these errors are considered, as well as the methods of their compensation. In our work, we consider the spectrometer as a “black box”, so we are not interested in the physical mechanisms of the spectrum distortion. We only operate with the data that it is capable of producing (taking into account the built-in mechanisms for picking up and converting the analog output of the CCD to the digital output) in response to standard commands–requests. It is also worth noting the work [27], where a modification of the algorithm for calculating the information characteristics of the FBG spectrum with its significant distortions is proposed.

Nevertheless, the interrogation system must consider the spectral characteristics both of spectrum analyzers and broadband light sources. It is necessary to calibrate spectrum analyzers and light sources jointly if we want to use their combination as an FBG interrogation system to avoid the FBG spectrum distortion due to different exposition times.

Thus, the task is to study the influence of the TINT parameter on distortions of the shape of the response spectrum of the FBG sensor, and advise methods to improve them.

## 3. Collecting of Initial Data

In order to solve this problem, a described experimental set-up was used. Here, the same fiber-optic sensor based on FBG was interrogated from two ends of the fiber. The optical line length connected to the fiber sensor at one end was equal to 10 m, and at the other end was equal to 10 km. The maximum amplitude of the spectral response from the sensor was set at ~40,000 quantization levels of the analog-to-digital converter to ensure an acceptable signal-to-noise ratio. This, in turn, led to the fact that the sensor was interrogated from the short end with TINT equal to 20 ms, and from the long end equal to 135 ms. The central wavelength was determined by the center of mass method for the same wavelength range of this sensor. It was found that the error in determining the central wavelength value of FBG in this experimental setup could reach 10 pm, which caused an error equal to ~1 K in temperature determination. Figure 1 shows the spectral range of FBG, measured at three different values of TINT, the ordinate axis is shown in convenient units, normed at 40,000 quantization levels. The used spectrometer (Ibsen I-MON) has a built-in temperature sensor, the data from which are used to compensate the temperature fluctuations of the spectrometer elements (optical and electronic paths) and associated errors in the interpretation of the FBG spectrum. The method of this compensation is described in the documentation on Ibsen I-MON.

The central wavelength value, calculated for the same spectrum of FBG at different TINT values, can differ significantly in practice. The difference in determining the FBG central wavelength depends on the chosen method and can reach 40 pm. For example, as shown in Figure 1, the differences between the central wavelength values for FBG at different values of TINT (20, 73, and 135 ms), are 7.6, 9.5, and 1.9 pm. This accuracy cannot be satisfactory, when the requirements for the temperature determining the accuracy is less than 1 K.

For further investigation of the hypothesis, the initial broadband spectra were obtained at different values of TINT on the spectrum analyzer. Figure 2 shows the spectral characteristics of broadband laser radiation, obtained at different values of TINT (curve 1–20 ms, 2–48 ms, 3–77 ms, 4–106 ms, 5–135 ms).

For eleven different characteristic points of the spectrum, the dependence of their amplitudes on TINT was received; the dependences for these points with different initial values of the amplitude on TINT are shown in different colors in Figure 3. The dependence of the amplitude on TINT was plotted for the TINT range from 20 to 135 ms with the step equal to 1 ms, which provided 116 measurements.

From the dependences shown in Figure 3, it can be seen that, with an increase in TINT, a linear increase in the amplitude occurs; however, the coefficient of this linear dependence (the slope of the straight line) itself depends on the amplitude initial value, which is measured at the initial TINT value. In addition, it can be seen that the higher the amplitude, which is measured at the initial TINT value (at 20 ms), the faster it grows with the TINT increase. This leads to nonlinear distortion of the spectrum and, ultimately, to errors in determining the FBG central wavelength.

The dependence of the slope angle on the initial value of the amplitude obtained at the initial value of TINT was received. This dependence is shown in Figure 4. As can be seen from Figure 4, the slope of the linear dependence has a linear dependence on the initial amplitude.

## 4. Mathematical Processing

The spectral response of the broadband source radiation for initial TINT value (*t*_0_) can be obtained:(1)$$A_{i,0},\;i=\bar{1,N},$$
where *N* is the number of sampling points of the spectrum, *i* = 1 and *i* = *N* correspond to minimal and maximum wavelengths of range, respectively.

The amplitude increases linearly with TINT increasing, which allows approximating the dependence by a straight line:(2)$$A(t)=A_{0}+k(A_{0})\cdot (t-t_{0}),$$
where *t*_0_ is the initial TINT value, at which the calibration is performed; *t* is the arbitrary TINT value; *k*(*A*_0_) is the linear dependence slope; and *A*_0_ is the amplitude, obtained at the initial TINT (*t*_0_).

Based on the fact that the slope of the linear dependence (Figure 4) also increases linearly with the increasing amplitude, measured at the initial TINT, the slope of the linear dependence in Equation (2) can be represented as its linear dependence on the initial value of the amplitude, obtained at initial TINT. This statement can be formulated as follows:(3)$$k(A_{0})=\alpha \cdot A_{0}+\beta .$$

Substituting Equation (3) into Equation (2), we obtain the dependence of the amplitude on TINT and the initial value of the amplitude, obtained at initial TINT value. A field of amplitude values depending on the TINT and the initial value of the amplitude, measured at initial TINT, can be obtained:(4)$$A(t)=A_{0}+(\alpha \cdot A_{0}+\beta )\cdot (t-t_{0}),$$
where α and β are the coefficients of the linear dependence, *t*_0_ is initial TINT, at which the amplitudes *A*_0_ are measured.

It is possible to obtain a field of amplitude values depending on TINT and the initial value of the amplitude, measured at the initial TINT value, for the entire spectrum of the broadband radiation source:(5)$$\{A_{i,j},t_{j}\},\;i=\bar{1,N},j=\bar{0,M},$$
where *N* is the number of the spectrum sampling points, *M* is the number of points for the TINT, changing in the range from *t*_0_ to *t*_M_, namely *t_j_* = *t*_0_ + j(*t*_M_ − *t*_0_)/*M*.

Therefore, the measured amplitude *A_i_*_,*j*_(*t_j_*,*A_i_*_,0_) for *i*-th sampling point can be written as dependence on TINT (*t_j_*) and on the initial amplitude value *A_i,_*_0_ for each:(6)$$A_{i,j}(t_{j},A_{i,0})=A_{i,0}+(t_{j}-t_{0})(\alpha \cdot A_{i,0}+\beta ),\;\;i=\bar{1,N},j=\bar{1,M}.$$

The measured field of values Equation (5) makes it possible to determine the value of the slope of the linear dependence *k*(*A_i_*_,0_) in Equation (2) for each initial value of the amplitude in each *i*-th spectrum point by calculating the characteristic points {*A_i_*_,0_, *k*(*A_i_*_,0_)}, *i* = 1,*N* using the least square method by formula:(7)$$K_{i}=k(A_{i,0})=\sum_{j=1}^{M}\left(A_{j,i}-A_{0,i})(t_{j}-t_{0}\right)/\sum_{j=1}^{N}{\left(t_{j}-t_{0}\right)}^{2},\;\;\;\;i=\bar{1,N}.$$

It gives a set of values {*A_i_*_,0_, *K_i_*}, *i* = 1, *N*, which, in turn, allow determining the α and β values–the linear dependence coefficients of the slope on *A*_0_ in Equation (3), by the linear equations system solving the following:(8)$$\left(\begin{array}{cc} \sum_{i=1}^{N}A_{i,0}^{2} & \sum_{i=1}^{N}A_{i,0}\\ \sum_{i=1}^{N}A_{i,0} & N \end{array}\right)\cdot \left(\begin{array}{c} \alpha \\ \beta \end{array}\right)=\left(\begin{array}{c} \sum_{i=1}^{N}A_{i,0}K_{i}\\ \sum_{i=1}^{N}K_{i} \end{array}\right).$$

Thus, after calculating the set of values {*A_i_*_,0_, *K_i_*}, *i* = 1, *N*, and the coefficients of the linear dependence of the slope in Equation (3)–α and β, in relation to Equation (6), the entire right-hand side becomes known. It allows to recalculate the initial amplitude of the arbitrary spectrum point, measured at initial TINT, depending on the amplitude, measured at arbitrary TINT (*t*):(9)$$A_{i,0}=\frac{A{\left(t,A_{i,0}\right)}_{i}-\beta (t-t_{0})}{1+\alpha \cdot (t-t_{0})}.$$

In Equation (9), *A*(*t*, *A_i_*_,0_) is the amplitude, measured at arbitrary TINT, α and β are the linear dependence coefficients of the slope in Equation (3), obtained as the solution of the linear equation system in Equation (8), and *A_i_*_,0_ is the amplitude, measured at initial TINT.

Thus, knowing the current value of the amplitude at a point in the spectrum, measured at an arbitrary TINT, providing a comfortable signal-to-noise ratio, allows determining the amplitude value, measured at initial TINT, which is used in the calibration process for this sensor.

## 5. Experimental Setup

The experimental setup is shown in Figure 5a. The experimental setup consists of the thermostat, interrogator (Figure 5b), fiber sensors under calibration, and controlling system. The thermostat is shown at the left side of Figure 5a; it has its own temperature sensor which is connected to the interrogator by the RS-232 interface. The sensors are placed in the thermostat and held at a constant temperature. The thermostat’s temperature is varied in the range with the given step. The interrogator is placed on the right side of the table in Figure 5a. The experimental setup is assembled so the sensors are affected only by temperature.

The FBG interrogator (Figure 5b) includes the optical spectrum analyzer IBSEN I-MON 512 with a USB interface–1. The laser source SLD-761-HP1-DIL–2 is used as the broadband optical light source. The optical channel switch Sercalo MEMS switch rSC 1 × 8–3 is used to switch channels to have a possibility to interrogate several optical channels consistently with the given interval (~100 ÷ 200 ms). In this interrogator model, we used eight independent channels with eight sensors in each channel. The optical light source with the channel switch is maintained by our own designed control board–4. All measured data are collected on the on-board computer, Wafer ULT-3–5. The optical cross–6 is used for internal optical cabling. The common power supply–7 is used.

## 6. Calibration Data

The calibration curves received without and with the spectrum calibration algorithm are presented in Figure 6. All measurements were made for the bore-type temperature sensor FS62WSS (HBM) from two ends. The measurements made from the first end are marked by red rhombuses, and from the second end are marked by blue rhombuses. All measurements were made in the temperature range from 15 to 90 °C with a discrete step of 5 °C. Fifty independent measurements of the FBG central wavelength were made in each temperature point; thus, each rhombus in the figure means one measurement.

As one can see, there is dispersion in obtaining the FBG central wavelength in the case when the spectrum calibration algorithm is not used (Figure 6a). The average error of the central wavelength approximation is 50 pm, which is equivalent to 5 K. While the calibration dependence received with the spectrum correction algorithm gives an error approximation of the FBG central wavelength of less than 0.1 pm, that allows approximating the temperature with 0.01 K accuracy (Figure 5b).

The received results approve the necessity of using the preliminary mandatory calibration of the light source with the CCD element and using these calibration results to correct the FBG spectra from the spectrum analyzer.

## 7. Conclusions

The proposed algorithm allows excluding nonlinear distortions of the fiber Bragg grating spectrum response, caused by different signal integration times on charge-coupled devices.

The spectrum correction, performed according to the proposed algorithm, allows excluding limitations associated with the fact that the calibration of all sensors in the fiber-optic measurement system, based on CCD, must be carried out at the same signal integration time values, which will be used in exploitation.

As a result, the compensation of nonlinear distortions allows calibrating sensors independently from the measuring system. In addition, it removes restrictions on strictly fixed optical lengths from an interrogator to the sensor. Moreover, it gives the opportunity to simplify the array of sensors formed in an optical channel, and to replace sensors with one another arbitrarily during exploitation.

We are free to discuss this problem and our results with scientists and manufacturers.

## Figures and Tables

**Figure 1 sensors-21-02817-f001:**
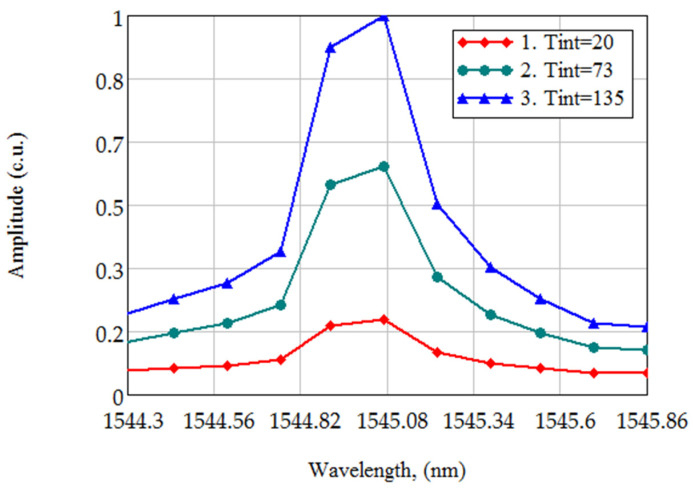
The FBG spectrum, measured at different TINT values (20, 73, 135 ms).

**Figure 2 sensors-21-02817-f002:**
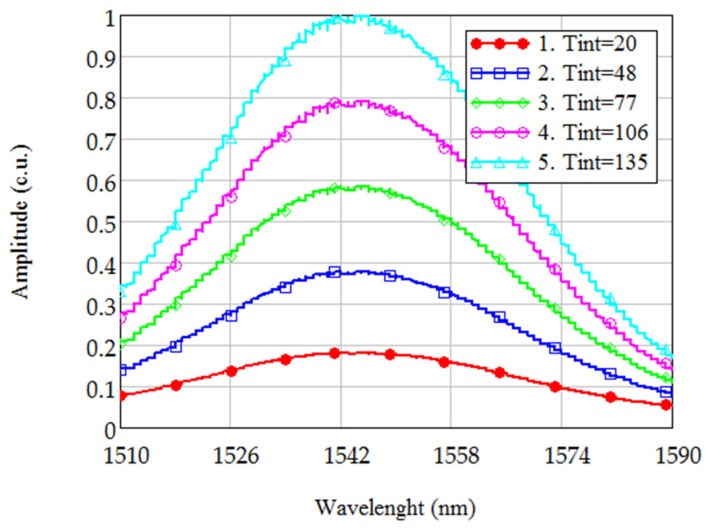
The broadband radiation spectra, obtained at different TINT values (1–20 ms, 2–48 ms, 3–77 ms, 4–106 ms, 5–135 ms).

**Figure 3 sensors-21-02817-f003:**
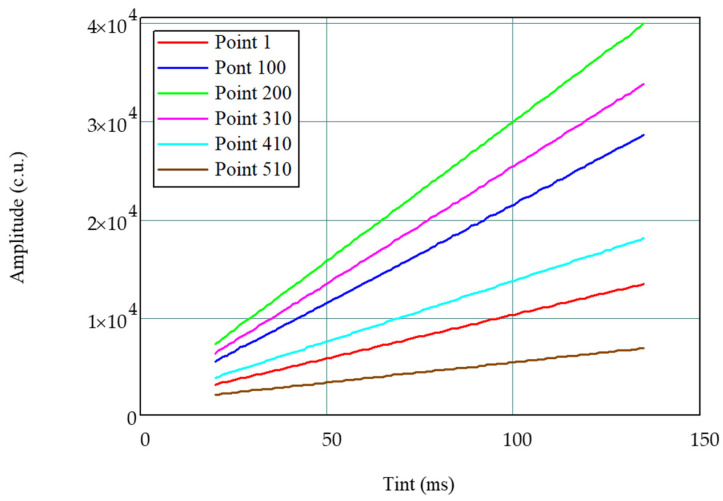
The dependence of the amplitude of different points of the spectrum on TINT.

**Figure 4 sensors-21-02817-f004:**
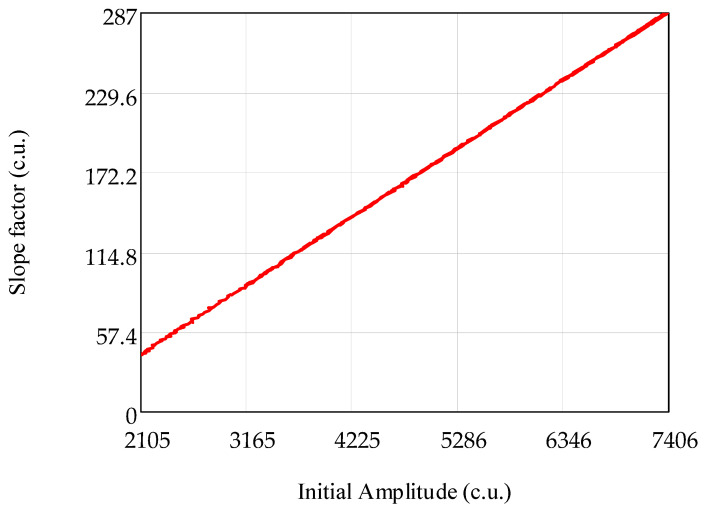
Dependence of the slope on the initial amplitude.

**Figure 5 sensors-21-02817-f005:**
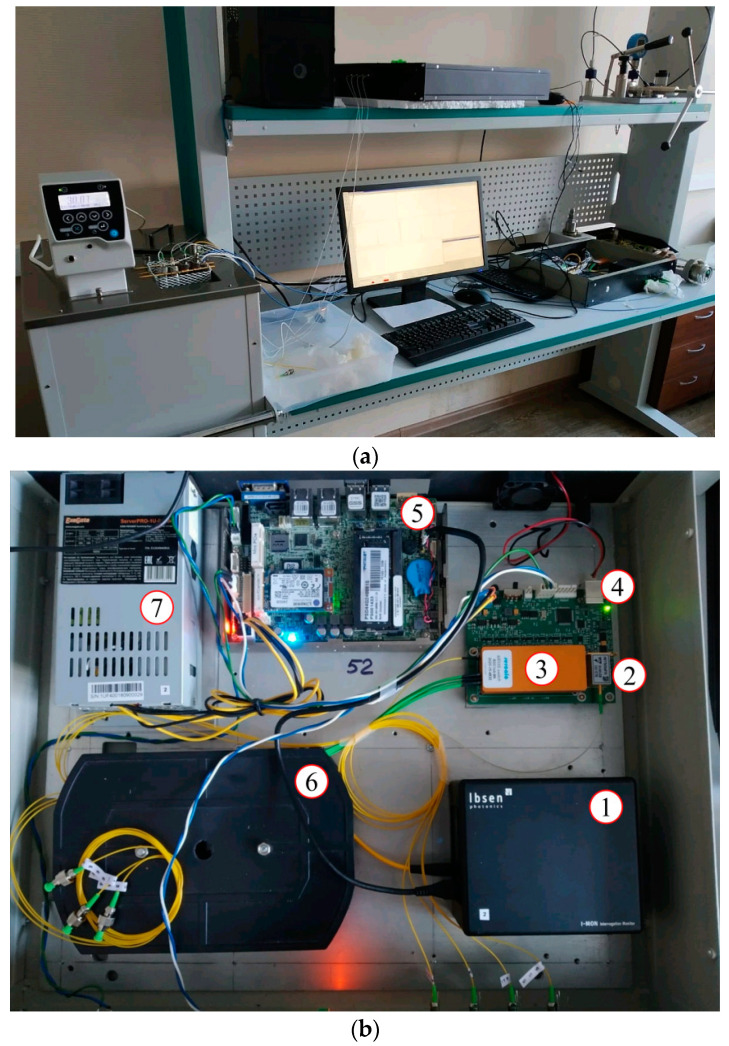
Experimental setup (**a**); FBG interrogator (**b**): 1–IBSEN I-MON 512 USB; 2–laser source SLD-761-HP1-DIL; 3–optic channel switch Sercalo MEMS switch rSC 1 × 8; 4–laser source and optical switch maintaining module; 5–computer Wafer ULT-3; 6–optical cross; 7–power supply.

**Figure 6 sensors-21-02817-f006:**
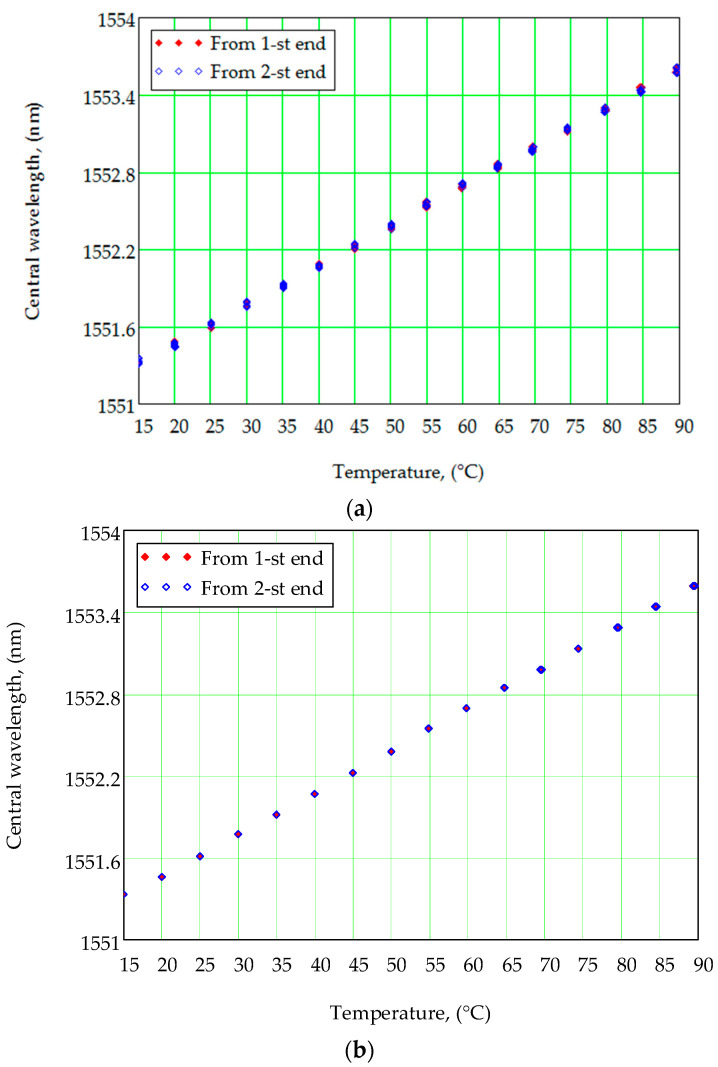
Calibration curves–FBG central wavelength dependences on temperature. The measured data: (**a**) without spectrum correction; (**b**) with spectrum correction.

## Data Availability

The data presented in this study are available on request from the corresponding author. The data are not publicly available due to rules of our contract conditions with our customer.

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
