# Peer review of "Algorithm of FBG Spectrum Distortion Correction for Optical Spectra Analyzers with CCD Elements"

_sensors, 2021, doi:10.3390/s21082817_

Round 1
Reviewer 1 Report
Dear authors, thank you for this article. However, the following improvements need to be made.
1) The problem of nonlinear distortion in CCD is related to the fact that the number of analogue-to-digital units is not proportional to the number of incident photons. This nonlinear phenomenon is characterized by a polynomial characteristic of up to 7 orders of magnitude. [1] In your article, you describe the distortion caused by a different TINT value. Please discuss in the article both distortion and compare their effects.
2) The graphical processing of the article must be improved. Graph labels are difficult to read.
3) The introduction does not provide sufficient background is necessary to include other relevant references oriented to nonlinear distortion and their description. For example [2]
4) Please provide a more detailed description of equations (1), and (5) to (9).
References:
[1] A non-linearity correction method of charge-coupled device array spectrometer
[2] The Study of Spectral Correction Algorithm of Charge-Coupled Device Array Spectrometer
[3] Spectral correction of the measurement CCD array
[4] Accurate wavelength calibration method for compact CCD spectrometer
Author Response
Dear authors, thank you for this article. However, the following improvements need to be made.
1) The problem of nonlinear distortion in CCD is related to the fact that the number of analog-to-digital units is not proportional to the number of incident photons. This nonlinear phenomenon is characterized by a polynomial characteristic of up to 7 orders of magnitude. [1] In your article, you describe the distortion caused by a different TINT value. Please discuss in the article both distortion and compare their effects.
We agree with the statement that CCD elements are nonlinearly sensitive to incident radiation and the link you provided was helpful in characterizing this phenomenon. In this work, we decided to solve a specific practical problem, leaving physical nuances in the background. On your recommendation, we have supplemented the content of the article (see the end of chapter 2).
2) The graphical processing of the article must be improved. Graph labels are difficult to read.
Done
3) The introduction does not provide sufficient background is necessary to include other relevant references oriented to nonlinear distortion and their description. For example [2]
Done, in agreement with 1.
4) Please provide a more detailed description of equations (1), and (5) to (9).
We have added some additional comments to the formulas. We would not like to give a complete mathematical background, firstly because it will not be difficult for readers of the magazine to repeat it, as it seems to us, and secondly, so as not to overload the article with formulas.
Reviewer 2 Report
This paper reports an fiber Bragg grating spectral correction algorithm for eliminating its nonlinear distortions which is caused by different signal integration time on charge-coupled devices. In general, the paper is well organized and the results are convincing. However, it requires moderate revisions before it can be accepted for publication. The comments are listed as follows:
- This proposed work focuses on the nonlinear distortions of the fiber Bragg grating spectrum, but the authors just used a little of space to talk about it in Section 1 “Introduction”. It is better to describe the previous works reported in more detail. For example, the following reference should be mentioned.
A Lamberti, S Vanlanduit, B De Pauw, F Berghmans, Influence of Fiber Bragg Grating Spectrum Degradation on the Performance of Sensor Interrogation Algorithms, Sensors 2014, 14, 24258-24277.
2. The authors pointed out that the nonlinear distortions of the fiber Bragg grating spectrum response was caused by different signal integration time on charge-coupled devices in Section 7 “Conclusions”, However, the nonlinear distortions may be caused by the central wavelength drift of the broadband laser. How can one distinguish them and eliminate the distortion interference resulted from the laser wavelength shift?
3. It’s better to list the error of central wavelength at each temperature in Figure 6 by using a table.
4. Please make the fonts bigger and the curves bolder in all thefigure, so that the readers can read them clearly.
5. The author should check carefully the grammar and text errors in the paper. Some examples are given:
--- the abbreviation of “signal exposition time (TINT)” may be not correct.
--- The meaning of this sentence “the radiation of which is received” (in line 34-35of page 1) is not clear.
--- The sentence (in line 218-219 of page 8) “it can lids to temperature ap-218 proximation error in 5 K”should be checked.
--- The sentence “For twelve different characteristic points of the spectrum” should be modified into “For eleven different characteristic points”, since only eleven points are contained in the Fig. 1.
Author Response
This paper reports a fiber Bragg grating spectral correction algorithm for eliminating its nonlinear distortions which is caused by different signal integration time on charge-coupled devices. In general, the paper is well organized and the results are convincing. However, it requires moderate revisions before it can be accepted for publication. The comments are listed as follows:
- This proposed work focuses on the nonlinear distortions of the fiber Bragg grating spectrum, but the authors just used a little space to talk about it in Section 1 “Introduction”. It is better to describe the previous works reported in more detail. For example, the following reference should be mentioned.
A Lamberti, S Vanlanduit, B De Pauw, F Berghmans, Influence of Fiber Bragg Grating Spectrum Degradation on the Performance of Sensor Interrogation Algorithms, Sensors 2014, 14, 24258-24277.
This article (Influence of Fiber Bragg Grating Spectrum Degradation on the Performance of Sensor Interrogation Algorithms) mostly describes probing FBG with distorted spectrum. In our experiments the FBG spectrum is not distorted itself, distortion was caused by implementation of interrogating device. But this research can be interesting in our spectrum post-processing algorithms. We also added this article to reference list.
- The authors pointed out that the nonlinear distortions of the fiber Bragg grating spectrum response was caused by different signal integration time on charge-coupled devices in Section 7 “Conclusions”, However, the nonlinear distortions may be caused by the central wavelength drift of the broadband laser. How can one distinguish them and eliminate the distortion interference resulted from the laser wavelength shift?
The stability of the spectrum of the laser diode is provided by a high-precision driver, which ensures the accuracy of maintaining the temperature at the level of 0.1 °Ð¡ and the current of 10 μA (0.1 % from nominal). The stability of the spectrum was checked by long-term measurement of the response, so the difference was of the order of the CCD intrinsic noise.
- It’s better to list the error of central wavelength at each temperature in Figure 6 by using a table.
Done
- Please make the fonts bigger and the curves bolder in all the figures, so that the readers can read them clearly.
Done
- The author should check carefully the grammar and text errors in the paper. Some examples are given:
- the abbreviation of “signal exposition time (TINT)” may be not correct.
The spectrometer manufacturer (Ibsen) uses this term (as variable and command in software). The term means “Time of INTegration”. This denotation was added to the text.
- The meaning of this sentence “the radiation of which is received” (in line 34-35of page 1) is not clear.
Done
- The sentence (in line 218-219 of page 8) “it can lids to temperature ap-218 proximation error in 5 K”should be checked.
Done
- The sentence “For twelve different characteristic points of the spectrum” should be modified into “For eleven different characteristic points”, since only eleven points are contained in the Fig. 1.
Done
Reviewer 3 Report
Authors presented a technique for spectral correction in FBGs. The main assumption is the exposition time influences the central wavelength, which makes sense. However, the technique needs to be further explanations and many clarifications listed below.
- The explanation and implementation were not clear enough. It seems that there is a linear relation between curve slope and exposition, authors should clarify if there is only a linear correction, based on figure 4 and further discuss the implementation as the calibration section is too brief, more experiments at different conditions of exposition times are needed.
- It is not possible to understand the lines at figure 3, it seems to represent different exposition times, but it is not very clear, especially the legends of each curve.
- The spectra at each exposition time condition should be presented after the application of the correction technique to verify their correction.
- Spectrometers generally suffer from temperature influence, even OSAs suffer from such influence. Thus, authors should provide the temperature inside the interrogation unit (shown in Figure 5b) to verify if this effect is due to minor temperature variations close to the spectrometer. In addition, it would be beneficial if the authors perform the experiments with temperature variations inside the interrogator. Of course, the temperature variation should be smaller than the one in the FBGs due to its smaller temperature range, but a 10°C-20°C variation would be important to verify if this effect is correlated to the temperature on the spectrometer.
- More experiments in Section 6 are needed, authors should test the technique under different exposition times, at different number of cycles (and ranges) and, if possible, with small temperature variations in the spectrometer region.
- English revision is needed, there are some grammar errors, confusing sentences and misplaced expressions.
Author Response
Authors presented a technique for spectral correction in FBGs. The main assumption is the exposition time influences the central wavelength, which makes sense. However, the technique needs to be further explanations and many clarifications listed below.
- The explanation and implementation were not clear enough. It seems that there is a linear relation between curve slope and exposition, authors should clarify if there is only a linear correction, based on figure 4 and further discuss the implementation as the calibration section is too brief, more experiments at different conditions of exposition times are needed.
The authors carried out a large series of experiments to reveal the dependence of the obtained spectral response for various FBGs and passive optical devices. The analysis showed that to determine the parameters of the distortion of our "black box", it is sufficient to measure the initial broadband radiation, considering its characteristics to be a reference, since it is this source that is used for illumination of the FBG. We could give graphs of all our preliminary measurements of various devices and hypotheses put forward in the process. At the same time, as it seems to us, this can only confuse the reader, so in the article, we have left only those materials that directly lead the reader to conclusions. We are ready in personal correspondence with you to reveal all the nuances that remained outside the scope of the manuscript.
- It is not possible to understand the lines at figure 3, it seems to represent different exposition times, but it is not very clear, especially the legends of each curve.
Done
- The spectra at each exposition time condition should be presented after the application of the correction technique to verify their correction.
Done
- Spectrometers generally suffer from temperature influence, even OSAs suffer from such influence. Thus, authors should provide the temperature inside the interrogation unit (shown in Figure 5b) to verify if this effect is due to minor temperature variations close to the spectrometer. In addition, it would be beneficial if the authors perform the experiments with temperature variations inside the interrogator. Of course, the temperature variation should be smaller than the one in the FBGs due to its smaller temperature range, but a 10°C-20°C variation would be important to verify if this effect is correlated to the temperature on the spectrometer.
The spectrometer, which was used in the experiments (Ibsen I-MON), has a built-in temperature sensor, the readings from which are used to compensate for the temperature fluctuations of the spectrometer elements (optical and electronic paths) and associated errors in the interpretation of the FBG spectrum. The method of this compensation is described in the documentation on Ibsen I-MON. The experiment you described, with both the active and inactive option of the built-in thermal compensation of the spectrometer, we will be ready to carry out as part of a more detailed study of this topic in the following works.
- More experiments in Section 6 are needed, authors should test the technique under different exposition times, at different number of cycles (and ranges) and, if possible, with small temperature variations in the spectrometer region.
In fact, we have carried out a lot of all kinds of experiments, with different types of FBGs, optical filters, multiplexers, etc. The experiments were carried out at different temperatures and the data were analyzed at different wavelengths within the I-MON measurement range. Each measurement of wavelength and spectra was made in the fast measure mode, which involves taking 128 spectra at a frequency of 2000 Hz in one measurement cycle (the measurement takes about 64 μs). The subsequent elimination of the obtained data makes it possible to eliminate the influence of the fluctuation of the measuring system on the measurement results. The resulting averaged spectrum is considered to be one spectrum measurement obtained for a given temperature at a given accumulation time.
- English revision is needed, there are some grammar errors, confusing sentences and misplaced expressions.
Done
Round 2
Reviewer 1 Report
Thank you for the corrections.
Reviewer 3 Report
Authors addressed most of my comments.